# Magnetic Resonance Imaging Findings Corresponding to Vasculitis as Defined via [^18^F]FDG Positron Emission Tomography or Ultrasound

**DOI:** 10.3390/diagnostics13233559

**Published:** 2023-11-29

**Authors:** Andrea K. Hemmig, Christof Rottenburger, Markus Aschwanden, Christoph T. Berger, Diego Kyburz, Maurice Pradella, Daniel Staub, Stephan Imfeld, Gregor Sommer, Thomas Daikeler

**Affiliations:** 1Department of Rheumatology, University Hospital Basel, 4031 Basel, Switzerland; 2Division of Nuclear Medicine, University Hospital Basel, 4031 Basel, Switzerland; 3Department of Angiology, University Hospital Basel, 4031 Basel, Switzerland; 4University Center for Immunology, University Hospital Basel, 4031 Basel, Switzerland; 5Department of Biomedicine, University of Basel, 4031 Basel, Switzerland; 6Department of Radiology, Clinic of Radiology and Nuclear Medicine, University Hospital Basel, 4031 Basel, Switzerland; 7Institute for Radiology and Nuclear Medicine, Hirslanden Klinik St. Anna, 6006 Lucerne, Switzerland

**Keywords:** giant cell arteritis, imaging, vasculitis, magnetic resonance imaging, ultrasonography

## Abstract

**Background:** We sought to investigate magnetic resonance imaging (MRI) parameters that correspond to vasculitis observed via [^18^F]FDG positron emission tomography/computed tomography (PET/CT) and ultrasound in patients with large-vessel giant cell arteritis (LV-GCA). **Methods:** We performed a cross-sectional analysis of patients diagnosed with LV-GCA. Patients were selected if MRI, PET/CT, and vascular ultrasound were performed at the time of LV-GCA diagnosis. Imaging findings in vessel segments (axillary segment per side, thoracic aorta) assessed using at least two methods were compared. Vessel wall thickening, oedema, and contrast agent enhancement were each assessed via MRI. **Results:** Twelve patients with newly diagnosed LV-GCA were included (seven females, 58%; median age 72.1, IQR 65.5–74.2 years). The MRI results showed mural thickening in 9/24 axillary artery segments. All but 1 segment showed concomitant oedema, and additional contrast enhancement was found in 3/9 segments. In total, 8 of these 9 segments corresponded to vasculitic findings in the respective segments as observed via PET/CT, and 2/9 corresponded to vasculitis in the respective ultrasound images. If MRI was performed more than 6 days after starting prednisone treatment, thickening and oedema were seen in only 1/24 segments, which was also pathologic according to ultrasound findings but not those obtained via PET/CT. Four patients had mural thickening, oedema, and contrast enhancement in the aorta, among whom three patients also had vasculitic findings observed via PET/CT. Isolated mural thickening in one patient corresponded to a negative PET/CT result. **Conclusions:** In the MRI results, mural thickening due to oedema corresponded to vasculitic PET/CT findings but not vasculitic ultrasound findings. The duration of steroid treatment may reduce the sensitivity of MRI.

## 1. Introduction

Giant cell arteritis (GCA) is the most common primary vasculitis affecting the elderly. Patients with large-vessel GCA (LV-GCA) may present with an isolated systemic inflammatory syndrome, and imaging is necessary to establish a diagnosis [1]. Different imaging techniques measure different characteristics of the vessel wall in patients with LV-GCA. Magnetic resonance imaging (MRI) is used to assess vessel wall morphology, oedema, and capillary leaking via contrast enhancement [2,3,4]. [^18^F]fluorodeoxyglucose (FDG) positron emission tomography/computed tomography (PET/CT) is used to measure metabolic activity via glucose uptake [5,6], while ultrasound is used to visualize wall thickening based on echogenicity by using sound waves and their echoes [7]. For PET/CT and ultrasound, qualitative and quantitative parameters defining vasculitis have been proposed and are widely used [8]. In contrast to its use for diagnosing cranial GCA [2], magnetic resonance imaging (MRI) is poorly standardised for the diagnostics of LV-GCA, and only few studies exist regarding the diagnosis of extracranial LV-GCA via MRI [9,10,11,12,13,14,15]. Furthermore, studies comparing MRI with other imaging modalities are sparse, and their authors have used different approaches for MRI interpretation. In a study comparing a visual, semi-quantitative four-point PET score with findings from CT-angiography and MR-angiography (MRA) observed in patients with GCA or Takayasu arteritis (TAK), an increasing PET score was associated with the synchronous presence of wall thickening [12]. In a study comparing MRA with FDG-PET in patients with GCA or TAK, vascular involvement determined via MRA was defined by the presence of either wall thickening, oedema, stenosis, occlusion, or aneurysms. The authors found an association between vessel wall oedema and thickening determined via MRA, with PET scans being consistent with active vasculitis. However, the agreement between MRA and PET regarding the extent of the disease was only 60% [13]. Yip et al. compared MRI to ultrasonography findings in patients with GCA and defined vasculitic changes observed via MRI as mural thickening using a four-point ranking scale. In this study, the agreement between ultrasound and MRI regarding disease extent was 72.1% [15].

The aim of this study was to identify which MRI parameters correspond to PET/CT- or ultrasound-defined vasculitic segments in patients with LV-GCA.

## 2. Materials and Methods

### 2.1. Patients and Setting

This study is a cross-sectional analysis of patients diagnosed with LV-GCA between January 2019 and March 2023 at the University Hospital Basel, Switzerland. Patients with suspected LV-GCA were routinely screened via ultrasound of the supra-aortic vessels, and PET/CT was performed to assess aortic involvement. This study included newly diagnosed GCA patients who were found to have large-vessel involvement upon conducting PET/CT and who underwent an additional MRI scan at diagnosis. All three imaging modalities were performed within 4 weeks of starting treatment. The final diagnosis of LV-GCA was made if (i) temporal artery biopsy was positive or (ii) at least 2 of 5 1990 American College of Rheumatology criteria were fulfilled in combination with PET/CT findings typical for large-vessel vasculitis (LVV) [16]. All patients provided written informed consent. Patient characteristics are summarized in Appendix A.

### 2.2. MRI Acquisition and Assessment

MRI examinations were performed using a 1.5 T clinical MRI system (Magnetom Avanto fit, Siemens Healthineers, Erlangen, Germany) using the sequence protocol listed in Appendix A. For contrast-enhanced imaging, a Gadolinium-based contrast agent (Gadobutrol (Gadovist^®^), Bayer AG, Leverkusen, Germany) was applied at the standard dose of 0,1 mmol Gd/kg of body weight. Pathological MRI findings (i.e., mural thickening, mural oedema, and late mural enhancement) were assessed by a board-certified radiologist (G.S., 8 years of experience) subspecialized in cardiovascular imaging and blinded with respect to the clinical data, PET/CT scan, and ultrasound results. Mural thickening was scored as follows: 0 (no mural thickening), 1 (mild mural thickening; 2–3 mm for aorta and 1–2 mm for the subclavian/axillary artery), or 2 (strong thickening; >3 mm for aorta and >2 mm for the subclavian/axillary artery). Mural oedema was subjectively scored as 0 (no mural oedema), 1 (mild mural oedema), or 2 (strong mural oedema) using T2w BLADE and Diffusion-weighted sequences. Mural contrast enhancement was subjectively scored as 0 (no mural enhancement), 1 (mild mural enhancement), or 2 (strong mural enhancement and/or perivascular enhancement) using static and dynamic T1w sequences pre- and post contrast (Appendix A).

### 2.3. PET/CT Image Acquisition and Assessment

PET/CT scanning was performed using a Siemens Biograph PET/CT mCT128 scanner (Siemens Healthcare, Erlangen, Germany). Patients were fasting for at least 6 h before intravenous injection of 5 MBq of FDG/kg body weight at median glycaemic levels below 10 mmol/L, and scans were obtained 1 h after injection as previously described [6]. The PET/CT scans were assessed by an experienced board-certified nuclear medicine specialist (C.R., 17 years of experience) blinded to the clinical and complementary imaging results. The degree of FDG uptake was quantified using the maximum standardized uptake value (SUV) of the vessel divided by the mean SUV of the liver. Findings positive for vasculitis were defined as artery/liver SUV ratio >1 for the subclavian/axillary segment and >1.3 for the aorta, as previously validated [6].

### 2.4. Ultrasound Examination and Assessment

Vascular ultrasound imaging was conducted using EPIQ 7 duplex devices with a linear 12–3 MHz and 18–5 MHz transducer (both from Philips, Best, The Netherlands). All ultrasound images were evaluated by an experienced angiologist (M.A., 30 years of experience) blinded to the clinical data, MRI, and PET/CT results. The subclavian/axillary segments were bilaterally categorised as ‘normal’ or ‘vasculitis’. Vasculitis was defined as circumferential homogenous hypoechoic wall thickening that was well-delineated towards the luminal side and not showing arteriosclerotic lesions, as previously described [17].

### 2.5. Comparison of MRI with PET/CT and Ultrasound

Due to varying anatomical definitions in the different imaging modalities used, we combined the analysis of the subclavian and axillary segments into one segment per side (axillary artery) to improve comparability between the imaging methods. The axillary segments are commonly affected in LV-GCA and are accessible for all three imaging modalities [18]. Additionally, we compared vascular findings of the thoracic aorta between MRI and PET/CT for the same patients.

### 2.6. Statistical Analysis

As this study was exploratory, the study sample was not calculated, and the analyses presented in this manuscript are descriptive. Continuous variables are expressed as medians with interquartile ranges (IQR). Categorical variables are presented as numbers with percentages. All statistical analyses were performed using R version 4.1.1 (2021-08-10) [19].

## 3. Results

### 3.1. Study Population

Twelve patients (seven females, 58%; median age of 72.1 years, IQR 65.5–74.2) with newly diagnosed LV-GCA were included in this study. The median erythrocyte sedimentation rate was 64 mm/h (IQR 39.5–78.3), and the median C-reactive protein level was 40.6 (IQR 12.5–101.7). The most common symptoms were headache, jaw claudication, and polymyalgia (42% each). MRI was performed at a median of 16 days (IQR 6.8–29.8), PET/CT was performed at a median of 7 days (IQR 3.8–11.3), and ultrasound imaging was performed at a median of 4 days (IQR 2.0–12.3) after glucocorticoid treatment initiation.

### 3.2. MRI Findings Compared to PET/CT and Ultrasound in the Axillary Segments

The MRI results showed mural thickening in 9/24 (37.5%) segments. Of these, eight segments had additional mural oedema, of which three segments also showed additional contrast agent enhancement. Eight of nine thickened segments viewed via MRI were classified as vasculitis through PET/CT. Only one segment showed thickening and oedema in the MRI results, but without corresponding findings from PET/CT (Patient 10, right segment, SUV artery/liver ratio = 0.9 [cut-off > 1]). Two of nine segments with thickening observed via MRI had congruent vasculitic findings in the ultrasound results (one with a negative PET/CT result), but seven segments with thickening in the MRI results were negative when viewed using ultrasound.

Five segments were classified as vasculitis via PET/CT and ultrasound, and another three segments were classified as such via ultrasound only, which were normal when viewed via MRI. For all these patients, MRI was performed more than 6 days after the start of prednisone treatment (Table 1). 

### 3.3. MRI Findings Compared to PET/CT in the Thoracic Aorta

Four of twelve patients showed mural thickening with concomitant oedema and contrast enhancement of the thoracic aorta when viewed using MRI. Of these, three patients had congruent vasculitic findings observed via PET/CT. Only one patient with mural thickening, oedema, and enhancement as determined via MRI was not classified as having vasculitis according to PET/CT but had an SUV artery/liver ratio of 1.21, which is just below the applied cut-off of 1.3 for vasculitis (Patient 10). One patient had isolated thickening according to MRI of the thoracic aorta, which was negative on the corresponding PET/CT scan (Patient 6 in Table 1). Figure 1 shows the PET/CT and MRI findings of Patient 3.

## 4. Discussion

To date, there are no standardised criteria for the diagnosis of LV-GCA via MRI [1]. Here, we aimed to identify parameters obtained via MRI corresponding to vasculitis by comparing MRI to PET/CT and ultrasound at the segment level for the most-often-in-LV-GCA-involved arteries, the axillary segment, and the thoracic aorta.

The presence of oedematous wall thickening upon conducting MRI corresponded to vasculitic findings according to PET/CT, whereas isolated vessel wall thickening not related to oedema was found in two segments only upon conducting MRI, with one of these not showing any FDG uptake according to PET/CT. Non-oedematous wall thickening has been shown to be an unspecific finding that may be seen in overt atherosclerosis [20] or in patients with cardiovascular risk factors as a surrogate marker of subclinical arteriosclerosis [21]. Contrast enhancement was less frequently found in the axillary segment and did not increase the yield of pathological findings upon conducting MRI in our study. If contrast agents could be avoided, the availability of MRI for patients with allergies and renal insufficiency would increase, while the examination time would be reduced.

We found a high accordance of oedema determined via MRI with FDG uptake as assessed via PET/CT, but not with ultrasound. Only 1/8 segments that were classified as vasculitis according to MRI and PET/CT were positive according to ultrasound. This suggests that MRI and PET/CT, on the one hand, and ultrasound, on the other, visualise different vessel wall pathologies. We have previously shown similar findings for PET/CT compared to ultrasound [22].

MRI results rarely showed vasculitic changes after glucocorticoid medication for more than one week, consistent with a previously published study [11]. Similarly, the sensitivity of PET/CT has been reported to decrease with duration of glucocorticoid treatment, while ultrasound pathologies in the larger arteries seem to be less affected by steroid treatment [15,23]. This may explain the large discrepancy of subclavian/axillary segments being normal according to MRI but showing signs of vasculitis according to ultrasound in patients after more than 6 days of steroid treatment. Furthermore, the lower spatial resolution of MRI may have influenced its sensitivity in comparison with ultrasound. However, it remains unclear why ultrasound did not show vasculitis in the majority of segments that were positive according to both PET/CT and MRI. Due to the study’s inclusion criteria (patients with LV-GCA according to PET/CT at diagnosis of GCA), we cannot draw conclusions about the diagnostic accuracy of the different techniques.

This exploratory study has limitations: first, the number of included patients was small, which precluded statistical analysis. Thus, only limited inferences can be made from this study. Second, glucocorticoid treatment started before imaging, which was on average the longest before MRI. This may have had disparate effects on the sensitivities of the different imaging modalities.

In conclusion, vessel wall thickening due to oedema according to MRI was the essential feature of active vasculitis in direct comparison with PET/CT, while contrast agent enhancement appeared to be redundant and was seen less frequently than oedema. Pathologic findings from MRI had a low agreement with vasculitic ultrasound findings, suggesting that the presentation of vasculitis as seen via ultrasound is different from the vasculitic features seen via MRI imaging. Our results support the use of a second imaging modality in cases of suspected GCA but with ambiguous clinical, laboratory, or histological findings.

## Figures and Tables

**Figure 1 diagnostics-13-03559-f001:**
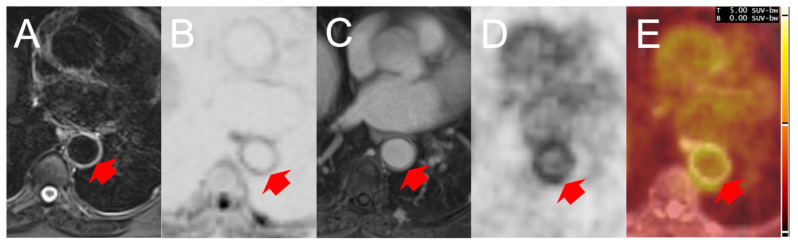
Findings rated positive for large-vessel vasculitis in the descending thoracic aorta according to MRI and FDG-PET/CT: MRI results show oedema on both T2w fat-suppressed (**A**) and low b-value diffusion-weighted sequences (**B**) accompanied by wall thickening and gadolinium enhancement according to post-contrast T1w GRASP (**C**). PET results show increased FDG uptake in the identical position (**D**). Figure (**E**) is a PET/MRI image fusion of the corresponding images (**B**,**D**) (Image fusion with Siemens SyngoVia version VB40). Red arrows highlight vessel wall oedema, contrast enhancement, and increased FDG uptake, respectively. Abbreviations: FDG = [^18^F]fluorodeoxyglucose; MRI = magnetic resonance imaging; PET/CT = FDG positron emission tomography/computed tomography; GRASP = Golden-angle radial sparse parallel (MRI).

**Table 1 diagnostics-13-03559-t001:** Imaging findings and duration of glucocorticoid treatment per patient in the axillary arteries and thoracic aorta.

	Axillary Artery						Thoracic Aorta				GC Treatment Days before Imaging		
Patient	Side	US	PET/CT	Mural MRI Findings			PET/CT	Mural MRI Findings			US	PET/CT	MRI
				Thickening	Oedema	C-Enhancement		Thickening	Oedema	C-Enhancement			
**1**	R	-	-	-	-	-	pos	strong	strong	strong	0	0	0
	L	-	pos	mild	mild	mild							
**2**	R	-	pos	-	-	-	pos	mild	mild	mild	0	0	0
	L	-	pos	mild	-	-							
**3**	R	pos	pos	mild	mild	-	pos	mild	mild	mild	0	0	4
	L	-	pos	mild	mild	-							
**4**	R	-	pos	mild	mild	-	pos	-	-	-	1	18	4
	L	-	pos	mild	mild	-							
**5**	R	-	pos	mild	strong	mild	-	-	-	-	2	4	6
	L	-	pos	mild	strong	mild							
**6**	R	pos	-	-	-	-	-	mild	-	-	3	3	9
	L	pos	-	-	-	-							
**7**	R	pos	pos	-	-	-	-	-	-	-	5	8	16
	L	pos	pos	-	-	-							
**8**	R	N/A	-	-	-	-	-	-	-	-	12	9	16
	L	-	-	-	-	-							
**9**	R	N/A	-	-	-	-	-	-	-	-	29	0	23
	L	pos	pos	-	-	-							
**10**	R	pos	-	mild	mild	-	-	strong	strong	mild	0	1	32
	L	pos	-	-	-	-							
**11**	R	pos	pos	-	-	-	pos	-	-	-	2	6	34
	L	pos	pos	-	-	-							
**12**	R	-	-	-	-	-	-	-	-	-	15	30	41
	L	-	-	-	-	-							

Abbreviations: GC = glucocorticoid; C-Enhancement = contrast enhancement; MRI = magnetic resonance imaging; PET/CT = [^18^F]fluorodeoxyglucose positron emission tomography/computed tomography; US = ultrasound; pos = positive; - = negative; N/A = not available.

## Data Availability

The data used and analysed during this study are available from the corresponding author upon reasonable request.

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
