# Peer review of "Magnetic Resonance Imaging Findings Corresponding to Vasculitis as Defined via [18F]FDG Positron Emission Tomography or Ultrasound"

_diagnostics, 2023, doi:10.3390/diagnostics13233559_

Round 1

Reviewer 1 Report

Comments and Suggestions for Authors

Magnetic resonance imaging findings corresponding to vasculitis as defined by [18F] FDG positron emission tomography or ultrasound.

This manuscript discusses an important and hot topic for the rheumatologists. However, some points need to be addressed.

Page

Line

Manuscript

Comment

1

18

 tomography (PET/CT) or ultrasound

The authors mean and not or

1

25

 MRI showed mural thickening in

The results should be arranged chronologically according to the methods

2

50

Furthermore, studies comparing MRI

It will be better of the authors give short notes about these studies and the nature of the comparison

2

62

PET/CT underwent a prospective thoracic  study MRI in the frame of our local Ethics

This is not a prospective ,this is still a cross sectional ..it is better to be edited as it is confusing

2

64

In addition, we included 4 patients from our local GCA cohort that un-64 derwent all three imaging modalities but were not prospectively included (Basler Riesen-65 zellarteriitis Kohorte [BARK EKNZ, #239/09]).

Un necessary details which may be confusing to the readers

3

100

Ultrasound examination and assessment

We need more idea about the operator and his /her experience and how the interpretation of the images done and if it depended on only one or two readers

3

113

2.6. Statistical analysis

More details are needed here

3

119

with newly 119 diagnosed LV-GCA

It is better to determine what is meant by “newly “and if the authors determine a specific period from the diagnosis

6

196

Conclusion

It is better to add the recommendations

Comments on the Quality of English Language

Moderate editing of English language required

Author Response

---------------------------------------------------------------

Reviewer 2 Report

Comments and Suggestions for Authors

Thank you for the opportunity to review this manuscript. In this study authors have analyzed the correlation between the findings of different imaging modalities in patients with GCA. This is an interesting study and has several strengths as well as limitations. Firstly, the analysis presented in this manuscript is purely descriptive. Without statistical analysis, only limited inferences can be made from this study. This should be mentioned in the limitation section. There are other limitations including the use of glucocorticoids prior to imaging that can affect the findings of imaging modalities is question, especially MRI. These have been limitations have been highlighted by the authors. The strength of this study is that both the radiologist and the nuclear medicine expert were blinded to the patients. Despite of the limitations of this study, it provides important preliminary evidence that can potentially provide guidance for future larger clinical studies. 

Comments on the Quality of English Language

The quality of English language is good with only minor errors present in the manuscript that can be addressed during editing.

Author Response

============================================

Round 2

Reviewer 1 Report

Comments and Suggestions for Authors

The authors addressed all comments